# Health and Culture: The Association between Healthcare Preferences for Non-Acute Conditions, Human Values and Social Norms

**DOI:** 10.3390/ijerph182312808

**Published:** 2021-12-04

**Authors:** Ingmar Leijen, Hester van Herk

**Affiliations:** School of Business and Economics, Vrije Universiteit Amsterdam, 1081 HV Amsterdam, The Netherlands; h.van.herk@vu.nl

**Keywords:** healthcare preferences, Schwartz values, tightness-looseness, European Social Survey

## Abstract

Preference for professional vs. non-professional or informal healthcare for non-acute medical situations influences healthcare use and varies strongly across countries. Important individual and country-level drivers of these preferences may be human values (the fundamental values that individuals hold and guide their behavior) and country-level characteristics such as social tightness (societal pressure for “acceptable” behavior). The aim of this study was to examine the relation of these individual and country-level characteristics with healthcare preferences. We examined European Social Survey data from 23,312 individuals in 16 European countries, using a multi-level, random effect approach, including individual and country-level factors. Healthcare preferences were explained by both human values (i.e., Schwartz values) and societal tightness (i.e., tightness-looseness scores by Gelfand). Stronger conservation increased, whereas self-transcendence and openness to change decreased preference for professional healthcare. In socially tight countries, we found a higher preference for professional healthcare. Furthermore, we found interactions between social tightness and human values. These results suggest that professional healthcare preference is related to both people’s values and societal tightness. This improved understanding is useful for both predicting and channeling healthcare seeking behavior across and within nations.

## 1. Introduction

Despite healthcare demand growing worldwide [1], there is still a knowledge gap regarding factors that drive peoples’ preferences for professional medical help. Economic and social consequences of seeking medical help in the area of non-acute diseases can be large as the prevalence of conditions such as backpain, headache, sleeplessness and a sore throat is high [2,3,4,5]. Although this preference can be influenced by factors such as accessibility and personal financial costs of the medical treatment, other factors can also play an important role. Moreover, dealing with limited resources, stakeholders in healthcare must outweigh costs and benefits when promoting healthcare seeking behaviors. On the one hand, a person’s decision to visit a doctor may lead to larger health costs than awaiting natural recovery or seeking help from friends or family. On the other hand, choosing not to visit a doctor could also lead to under-diagnoses of serious illnesses, threatening individual well-being and affecting society in the form of lost productivity. Thus, individuals, as well as medical professionals, healthcare organizations, insurance companies, health authorities and other stakeholders involved in the distribution of healthcare, will be in favor of self-help behavior for some conditions (e.g., to prevent antibiotics misuse), whereas for other conditions they may endorse professional help (e.g., vaccination or prevention programs). Despite the importance of guiding people towards either a medical professional or seeking self-help, insight into the drivers of the choice between these alternatives is scant. Therefore, healthcare preferences are, both in theory and practice, an important research area [6,7].

For identifying determinants for help-seeking behaviors, non-acute medical conditions (e.g., headache, backache, sore throat and sleeping problems) are particularly relevant, since people may have multiple options to choose from. These may vary from visiting a specialized clinic, a public health facility or general practitioner, or search other, non-professional help, such as asking help from family/friends, visiting a drugstore, calling a medical helpline or finding information online. Some people may decide to wait for natural recovery.

Existing research on healthcare preferences of people with non-acute conditions is mainly focused on the choice between emergency versus primary care [8,9]. These studies included socio-demographics and variables related to the healthcare service such as trust, convenience and familiarity [10,11]. Moreover, these single-country studies did not compare preferences between countries, whereas it is known there are sizable differences in healthcare behavior between nations.

A previous multi-country study looking at country characteristics and healthcare outcomes and behaviors focused on aggregate group-level data. For instance, smoking and drinking, and prevalence of diseases have been explained by country characteristics such as the state of healthcare, and national culture [12]. Others studied differences between specific cultural groups within society, such as ethnicity [13] to show differences in specific healthcare choice behavior. Although these studies show there are important differences between cultural groups in healthcare behavior, little is known about the factors related to healthcare preferences, and more specifically, preference for professional medical help by individuals from different countries.

In previous studies, several objective factors have been linked to healthcare preference and utilization, such as socio-demographics characteristics (such as income, education, and age) and context-related factors (such as characteristics of the national healthcare system (including insurance and healthcare accessibility) [14,15].

Other studies focused on socio-cultural factors influencing an individual’s choice for a certain healthcare option providing medical help such as trust in medical doctor, causing both satisfaction and loyalty [16]. Trust in medical doctors has been included in many studies [17,18], and a consistent finding is that it is an important factor in choosing professional medical help [19]. Thus, in addition to socio-demographics, it is expected that trust in medical doctor is an important factor related to preference for professional medical help.

An additional explanation might be related to what people consider important in their lives and drives their attitudes and behaviors; that is, the basic human values people hold [20]. People’s values relate to motivations, attitudes and behavior within the larger context of society [21,22,23], and thus have a profound influence on many daily life decisions.

The values people hold have been shown to explain or predict their behavior. The currently dominant theory that captures human values is the seminal theory by Schwartz [24]. In Schwartz’ theory, ten basic values are distinguished: benevolence, universalism, self-direction, stimulation, hedonism, achievement, power, security, conformity and tradition. Studies have combined these values into four higher order domains: openness to change (containing stimulation, self-direction, and hedonism), conservation (containing security, conformity, and tradition), self-enhancement (containing achievement, power) and self-transcendence (containing universalism, benevolence) [25,26], as visualized in Figure 1.

The combining of value types into the higher order value domains can be explained as follows: Conservation implies that a person who feels that conforming to norms set by society and family is important, as well as valuing personal security and traditions. Openness to change means that a person is not afraid of trying novel things, strives for creativity, and is not afraid of challenges that will foster mastery and independence. People who prefer conservation to openness to change values will prefer to behave in a way congruent with common behavior within one’s community. Individuals considering conservation values, (e.g., security), more important in their lives might reduce their perceived health risks by showing a higher preference for professional medical help to mitigate this risk. In contrast, people considering openness to change values more important, which implies a personal disposition more directed towards making independent choices, might have a higher propensity to take risks [23]. Furthermore, being more self-directed promotes entrepreneurial behavior [27] and is also associated with lower levels of anxiety [28]. Thus, we expect openness to change to increase the motivation for self-help, causing lower preference for professional medical help.

Self-enhancement, including power and achievement values, implies emphasizing the promotion of the self, prestige, social power, and control over others [24]. Since individuals with high self-enhancement focus more on themselves and their own benefit, and less on other people’s interests, we expect them to care strongly about the protection of their personal health and prefer care from a specialized medical professional. The opposing value of self-transcendence implies people to be more sensitive to the situation of others. This could also relate to a higher level of altruism and trust in (close) others [24]. Consequently, these individuals are expected to show a higher preference for seeking care from (close) others within their social environment, foregoing the option of visiting a professional medical doctor or nurse.

In addition to individual factors influencing health care preferences, societal factors may also play a vital role. However, little research has investigated the role of social norms within a society in the context of healthcare [29,30].

Given the expected importance of social norms in case of non-acute medical situations, we consider a country’s tightness-looseness as an important contributor to healthcare preference, as it specifically relates to the strength of social norms and tolerance of deviant behavior in society [31]. According to Gelfand’s conceptualization of tightness-looseness, in some nations, deviant behavior in public is more acceptable then in others; for instance, eating, arguing, kissing, singing, or listening to music in public is acceptable in some, but not in other nations. That is, in a tight society people will more likely have a higher need to show constrained behaviors, and will be high on self-control [31], feeling the need to comply with social norms in order to avoid disapproval. In contrast, in a loose society, deviation from the norms is more acceptable.

The presence or absence of pressure to do things in a specific way could influence the dependence on professional medical help; when the social norm is to seek professional help, it may lead to a higher preference for professional help in tight societies. Moreover, as societal tightness is considered a societal adaptation to survival in harsh circumstances [32], it may also be associated with a stronger focus on avoiding risk (in general) in that nation. In the latter context, visiting a healthcare specialist can be considered a risk reducing strategy. Thus, we expect that preference for choosing professional medical help will be higher in tight societies.

Summarizing, research investigating human preferences for professional medical help is scant, and mostly limited to emergency room (mis)use. Research on seeking help in non-acute medical conditions is as far as the authors are aware lacking. Especially in non-acute healthcare conditions we expect cultural context and human values to have a prominent role in determining healthcare preferences. Our contribution to the literature is, therefore, threefold. (1) Using a large dataset with 23,312 individuals from 16 countries we extend the literature on healthcare choice behavior by looking at preferences for professional medical help in non-acute medical conditions; (2) we investigate the relation of human values with the preference for professional healthcare in a non-acute medical condition; (3) we assess whether societal tightness-looseness inhibits or strengthens these relations.

## 2. Materials and Methods

### 2.1. Sample and Approach

Our data consisted of representative samples from the 2004 round of the European Social Survey (ESS). The ESS is a large scale, bi-annual, pan-European survey measuring social attitudes and behavior in nationally representative samples in more than 30 European countries. Data of this high-quality international survey is open source. For detailed information about this survey see [33]. We used the 2004 round of the ESS as it includes a module on health-related behavior. The ESS includes nations with more- and less developed economies and healthcare systems. For our research, we used respondents from the 16 countries for which tightness-looseness (TL) scores were available [31] Austria (*n* = 1644), Belgium (*n* = 1541), Germany (*n* = 2221), Estonia (*n* = 1140), Spain (*n* = 1340), France (*n* = 1499), United-Kingdom (*n* = 1593), Greece (*n* = 2071), Hungary (*n* = 1087), Iceland (*n* = 391), Netherlands (*n* = 1546), Norway (*n* = 1458), Poland (*n* = 1294), Portugal (*n* = 1741), Turkey (*n* = 1375), and Ukraine (*n* = 1371). We removed respondents under 18 and respondents with missing values on variables of interest, leading to a total sample size of 23,312 respondents (=78% of the original sample). The percentage of retained respondents varies between 57% for Estonia and 87% for Belgium.

### 2.2. Measures

From the ESS data, we included human values and used several control variables including trust and socio-demographic information of all individuals. At the country-level we included the variable tightness-looseness. Tightness-looseness values were taken from the research of Gelfand et al. [31]. We calculated preference for professional medical help by using a summated score of preference with four non-acute medical conditions: headache, backache, sore throat, and sleeping problems. For each problem one option for medical help could be selected: Doctor, nurse, nobody, friends or family, pharmacist/chemist/drug store, Internet/web, medical helpline, or another practitioner. When doctor or nurse was chosen, we allocated a score of “1” to our main dependent variable (i.e., preference for professional medical help), all other options received “0”. The choice for doctor or nurse varied between 42.1% (sore throat) and 77.8% (backpain). We added nurse to the choice for professional help (i.e., coded as 1) as in some countries nurses seem to have a more prominent role in giving care. We defined the difference based on payment for service: i.e., having a paid professional medical service versus all other non-paid options. We grouped pharmacist to the second option as this option could be seen as a way of self-help without paid consultation. The variable preference for professional medical help consisted of summating scores for the four non-acute medical conditions, resulting in a variable ranging from 0 to 4.

Schwartz human values were measured using the Portrait Value Questionnaire (PVQ-21) [33]. Human values were ipsatized following the common suggested procedure for analyzing values [33] and subsequently aggregated into the four higher-order value domains: openness to change, conservation, self-enhancement, and self-transcendence. Openness to change was calculated as the average of the values self-direction, stimulation, and hedonism, conservation was calculated as the average of conformity, tradition, and security. Self-enhancement was calculated as the average of achievement and power, and self-transcendence was calculated as the average of universalism and benevolence (see Figure 1).

To measure trust in the doctor, we developed a formative scale using items available in the ESS: we combined answers on 3 questions that captured several distinct aspects of trust in medical doctor. These questions measured equality, openness in communication and approachability of a doctor. Following established practice in cross-national research measurement [34], invariance for trust in medical doctor was assessed. Although the Chi-square was significant (large *n*) the other fit measures indicate a reasonable fit comparative Fit Index (CFI) = 0.957, Tucker–Lewis Index (TLI) = 0.933 and root mean square error of approximation (RMSEA) = 0.077). Thus, the scale was found to be metric invariant and can be used for further analysis in our multi-level regression analyses.

Several other control variables were included: At the individual level, interpersonal trust was measured with a 3-item, 11-point, reflective scale (Cronbach Alpha (α) = 0.765, min. α = 0.625 (Belgium), maximum α = 0.825 (Greece)). Trust in institutions was measured with a 3-item, 11-point reflective scale (α = 0.793, min. α = 0.726 (Poland), max. α = 0.825 (Hungary)). Following methodological standards [35], for both measures we assessed whether the scales were metric invariant across the 16 nations. Results showed that, although the chi-square was significant (as expected given the large *n*), the other fit indices indicated a good fit; for both measures CFI > 0.99, TLI > 0.98 and RMSEA < 0.06). Self-perceived-health (5-point scale) and perceived state-of-healthcare (11-point scale) were both measured with one item. Health impairment and having children in household (yes/no) were measured with one item. Gender was measured as 1 = male 0 = female, and age was measured in years. Education was measured using education in years; a measure in years that is highly correlated (r = 0.895, *p* < 0.001) with the harmonized ISLED (International Standard Level of Education) score to measure education [36].

### 2.3. Participants

Our sample was on average 46.9 years old (SD = 17.479; range 18 to 99, 54.6% female, 45.4% male). The average preference for professional medical help was 2.64 (SD = 1.42) with lowest average for Ukraine (1.68) and highest average for Turkey (3.39). We found significant differences in preference between nations (*F*_(15, 23,286)_ = 135.90, *p* < 0.001). In Figure 2, the preference for medical help in non-acute situations across nations is visualized.

For descriptive statistics of all main variables see Table 1. Correlations between variables in the model can be found in Appendix A (individual level Table A1, country level Table A2).

## 3. Results

### 3.1. Study Design

As individuals are nested in countries, we employed a multilevel regression approach [37,38] in the analysis of our data, investigating both individual (level 1) and national (level 2) effects. This approach is preferred in cross-national research as variance within nations often is much higher than variance between nations [25] there is a need to include both individual- and national-level characteristics to explain preference for medical help in non-acute medical conditions.

### 3.2. Multi-Level Analyses

In the next section we describe several nested multilevel regression models in which we present the additional effect of individual level human values and national level tightness-looseness on the preference for professional medical help in case of non-acute situations.

As common in multilevel regression modeling, we started with a null model, without any explanatory variables; this is the reference model. To test our model, we estimate a first model including socio-demographics and control variables. After estimating this first model, we the higher-order human values are included one by one. Values were added separately as they have high intercorrelations. As a next step, we add tightness-looseness at the country level, and subsequently the cross-level interaction between human values and societal tightness-looseness.

The null model, with random intercept for country and a random error term for individuals, showed that the variance in the data at the individual level is 1.776, and at the country level 0.129. This results in an intra-class correlation (*ICC*) of 0.087, meaning 8.7% of variance in the dependent variable was at the national level. As more than 5% of variance was shared by the people in one country, there was sufficient reason to warrant a multi-level-analysis approach [34]. To enable estimating added explained variance between our nested models, we used a Full-Maximum-Log-Likelihood (FML) approach as needed when nested models are to be compared [37]. Chi-Square difference tests (using −2 Log Likelihood) were used to assess significant changes in explained variance between respective nested models. To obtain robust confidence-intervals for all estimates we used a bootstrapping procedure (1000 iterations). Analyses were carried out using IBM SPSS 24.0 for multilevel modeling, and the package lavaan [39] in R [40] for metric invariance.

#### 3.2.1. Control Variables: Socio-Demographics, Attitudes, and Trust

In Table 2, we report five nested multilevel regression models. Model 1 is the model with only individual level variables. In Models 2 to 5, country-level tightness-looseness is added as well as the human values one at a time and the interaction term. In Model 1, the individual-level variables socio-demographics, attitudes, and trust are added as control variables and this results in a significant change (−2 Log Likelihood) in explained variance: (χ^2^_diff(11)_ = 756.58, *p* < 0.001), compared with the null model. In Model 1, a positive effect for age was found (*γ_AGE_* = 0.010, *p* < 0.001); The negative estimate for age-squared (*γ_AGE_^2^* = −0.0002, *p* <.001) indicates that the middle age group had a relatively higher preference for seeking professional help. Education had a significant negative effect (*γ_EDU_* = −0.011, *p* < 0.0001). Being health-impaired had a positive effect on professional medical help preference (*γ_HIMP_* = 0.052, *p* < 0.01) and women showed a higher preference for professional healthcare than men (*γ_GENDER_* = −0.128, *p* < 0.001). Finally, having children in a household increased preference for professional medical help: (*γ_CHILD_* = 0.031, *p* < 0.01). We also include the effect of attitudinal measures. The effect of the people’s perception of the state of healthcare services in one’s country is not significant: (*γ_HCS_* = −0.001, *p* > 0.05), whereas self-perceived health has a decreasing effect on preference (*γ_HEALTH_* = −0.050, *p* < 0.001). Interpersonal trust decreased preference for professional medical help (*γ_IPTR_* = −0.019, *p* < 0.001) whereas institutional trust has a significant positive effect on this preference (*γ_INTR_* = 0.029, *p* < 0.001). Finally, trust in medical doctor has a significant positive effect (*γ_MEDTRUST_* = 0.065, *p* < 0.001).

#### 3.2.2. Schwartz Values

In Models 2–5, we added the Schwartz values of Conservation, Openness to Change, Self-Enhancement and Self-Transcendence, respectively. Note that in Table 2, only the final models including effects of two levels and the cross-level effects are shown (models without country-level effects or interaction effects are available from the authors upon request). The reporting of only final models is justified, as the coefficients of the individual level effects did not change meaningfully when adding the country level variable tightness-looseness (TL) and cross level effects.

Conservation showed a significant positive relation with preference for professional medical help (*γ_CON_* = 0.102, *p* < 0.001) whereas openness to change had a significant negative relation (*γ_OTC_* = −0.064, *p* < 0.01). This opposite coefficient was expected as the two higher order values are opposites on the same bipolar value dimension.

Self-enhancement is not significant (*γ_SE_* = 0.007, *p* > 0.05), but self-transcendence decreased preference for professional medical help (*γ_ST_* = −0.049, *p* < 0.05)

#### 3.2.3. Country-Level Effect: Tightness-Looseness

The country-level variable tightness-looseness added to explained variance in all 4 multilevel regression models. Tightness-looseness statistically significantly contributed to explained variance in all four models. Compared with Model 1, Model 2_(cons)_: (*χ^2^_diff(3)_* = 110.78, *p* < 0.001), Model 3_(otc)_: (χ^2^*_diff(3)_* = 35.13, *p* < 0.001), Model 4_(se)_: (*χ^2^_diff(3_*_)_ = 16.64, *p* < 0.01), Model 5_(st)_: (*χ^2^_diff(3)_* = 41.49, *p* < 0.001). Societal tightness increases preference for professional medical help (*γ_TL_* between 0.129 and 0.147, all *p* < 0.01).

In Figure 3, a visualization of the relation of the country level tightness-looseness with average preference for professional medical help is provided. The figure suggests a linear relation between country level tightness-looseness and preference for professional medical help. This relationship seems unrelated to any north-south or east-west dichotomy, nor seems to have a direct relation to GDP-per-capita. For example, both in Norway with a high GDP-per-capita, and in Turkey with a low GDP-per-capita a high preference for professional medical help is found. Thus, there seems no clear pattern that could hint at such a simple geographical or economic explanation for relation between tightness-looseness and preference.

#### 3.2.4. Cross-Level Interactions between Trust, Values, and Tightness-Looseness

Finally, we estimated cross level interaction between (1) tightness-looseness and (2) the 4 higher order values conservation, openness to change, self-enhancement, and self-transcendence, respectively (Table 2). Results show that the interaction between tightness-looseness and conservation had an enhancing effect on preference for professional medical help (γ_CONS_ * TL = 0.025, *p* < 0.01); the effect of conservation is significantly stronger when societal tightness is higher. Furthermore, the interaction between tightness and self-transcendence was negative and significant (γ_ST_ * TL = −0.021, *p* < 0.01), showing that in more tight societies the negative effect of self-transcendence becomes stronger. Self-enhancement also showed a significant interaction effect, but the interaction did not add to explained variance of the model. Openness to change showed no significant interactions with tightness-looseness.

#### 3.2.5. Robustness-Checks

To assess robustness of our findings we performed several additional analyses. We considered the effect of adding a country level control variable (GPD-per-capita.) as well as the effect of substituting, tightness-looseness with GDP-per-capita, out-of-pocket healthcare expenditure, physician-density, and healthcare-insurance-systems. Information on physician density and GDP per capita in 2004 were taken from OECD [1] and UNDP [41]. To check whether a more privately versus a more governmentally financed healthcare structure would be affecting preferences, we created a measure that captured this dichotomy. Specifically, we coded healthcare-insurance systems within countries on a 4 point scale using data from KPMG [42] ranging from fully publicly financed (1), mainly publicly financed (2), mixed financed (3) to mainly privately financed (4). We observed no significant effects of any these variables. The estimates for tightness-looseness were stable when separately including each of the control variables on the country level.

We also took alternative other national-cultural dimensions to explain preference for professional medical help. We used Hofstede’s culture dimensions [43], which have been used in healthcare research before [12,44]. None of the main Hofstede dimensions (Individualism, Uncertainty avoidance, Masculinity and Power Distance) showed significant correlations with preference for professional medical help.

## 4. Discussion

Our study examined the relation of human values and societal tightness with the preference for professional medical in case of non-acute medical conditions as opposed to informal health care, controlling for socio-demographics and several factors such as trust. We found that both human values as well as societal tightness were related to preferences for medical healthcare.

First, we found that human values can help explain why people choose for professional medical help in non-acute situations. Conservation (being a measure of tradition, conformation to others and opposition to change) was positively related to preference for professional medical help. The negative relation of openness to change with preference is congruent with the positive relation of conservation. From these results we can conclude that preferring to obtain professional medical help is associated with a stronger importance attached to conservation values and less importance attached to openness values. Self-transcendence, measuring the degree to which people are more inclined to the need of others, was related to a decreasing preference for professional medical help. It may be assumed that valuing other people’s interests (versus valuing the self) is negatively associated with the preference for help from medical professionals.

Second, we showed that tightness-looseness [31] is an important predictor of preference for professional medical help in non-acute situations across countries. In additional analyses, in which we included several supply effects in our model, such as physician density, out-of-pocket expenditures, insurance systems, and GDP per capita, we still found that tightness was positively related to preference for professional medical help in non-acute conditions. Additionally, alternative cultural measures (i.e., Hofstede dimensions) did not show significant effects, adding to the robustness of the results. We found that the tighter a society, the more prone people will be to visit a doctor. Future research could look at the influence of tightness-looseness in other areas of healthcare behavior such as vaccination behavior/hesitation or emergency room (mis)use. Additionally, from a theoretical perspective, the interaction of individuals’ values with aspects of the environment they live in is an important avenue for further research.

Third, examining interaction effects, we found that social tightness is strengthening the positive relation of conservation with preference for professional medical help. We also found negative interaction of tightness with self-transcendence, suggesting that in a looser society the relation of self-transcendence with preference becomes stronger. This may make sense, as tightness-looseness is associated with a higher prevention-focus [45]: as a measure of freedom to deviate from norms and rules within society, it may also enhance or inhibit the relation of values with the preference for a medical professional in case of non-acute medical situations.

Taken together, our results show the relation of both societal context and human values with preference for professional medical help in the case of non-acute medical conditions. High scores on conservation values were related to stronger preference, and high scores on openness-to-change values were related to a weaker preference for help by a medical professional. High scores for tightness-looseness were related to a higher preference, and the effect of conservation was enhanced by tightness-looseness.

As in any study, our study has limitations: we were confined to a limited sample of 16 European countries, as we used existing data from both the European Social Survey and the published scores for tightness. Additionally, we could not directly account for cost of access to healthcare services, which might also be a predictor of preference. Nevertheless, we found with this limited number of countries that a substantial amount of variance on the aggregated level could be explained by tightness-looseness. Having a broader set of countries, with a more global scope, or looking at regional differences within countries could help in determining the boundaries of our findings.

## 5. Conclusions

Our results imply that, next to trust in medical doctor, both country level tightness-looseness and individual human values play an important role in the preference for seeking professional medical help. Higher importance attached to the Schwartz value domains conservation and the self-enhancement was associated with higher preferences for professional medical help, and also countries with higher levels of societal tightness faced a higher inclination to turn to professional medical help in case of non-acute medical situations. Our results may be of interest to governments, insurance companies, public policy makers, and general practitioners in delivering better attuned and personalized health care, improving communication to customers, or to increase or decrease care-seeking. As these common medical conditions are on the one hand related to increased cost of healthcare, and on the other hand to lost productivity and other social costs it is important to know that values add to explained variance in healthcare preferences. People who value conservation will be more prone to consult a medical professional, whereas on the other hand, people who value openness and self-transcendence will be less inclined to visit a doctor. To convince the latter groups of people to either use more, or less, professional medical services it is important to develop communication congruent with their values: focused messages appealing to values that people find important can strongly improve the effectivity of a communication strategy [46].

Lastly, general practitioners could use our results to improve communication and treatment of patients, as they can better predict the preference for going to a doctor with non-acute complaints, taking the cultural background of patients into account. This could be an informative cue to improve on communication about treatments.

## Figures and Tables

**Figure 1 ijerph-18-12808-f001:**
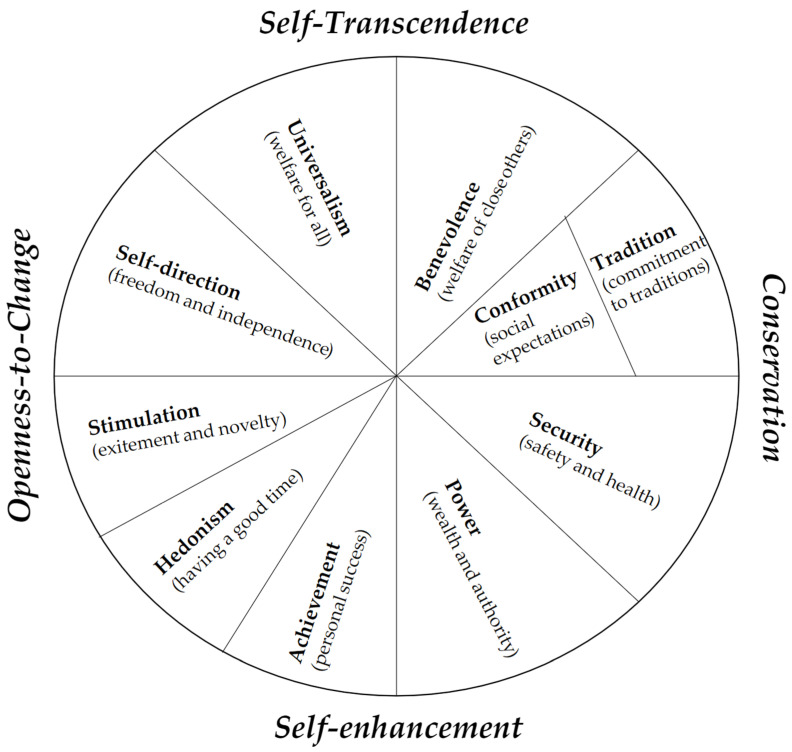
Human Values Framework, reprinted from Schwartz [24] with permission from Elsevier. The circumplex structure shows the compatibilities between adjacent values (e.g., universalism and benevolence) and oppositions between conflicting values (e.g., universalism and power). The labels outside the circumplex refer to combinations of values that are seen as higher order values (e.g., openness to change and conservation).

**Figure 2 ijerph-18-12808-f002:**
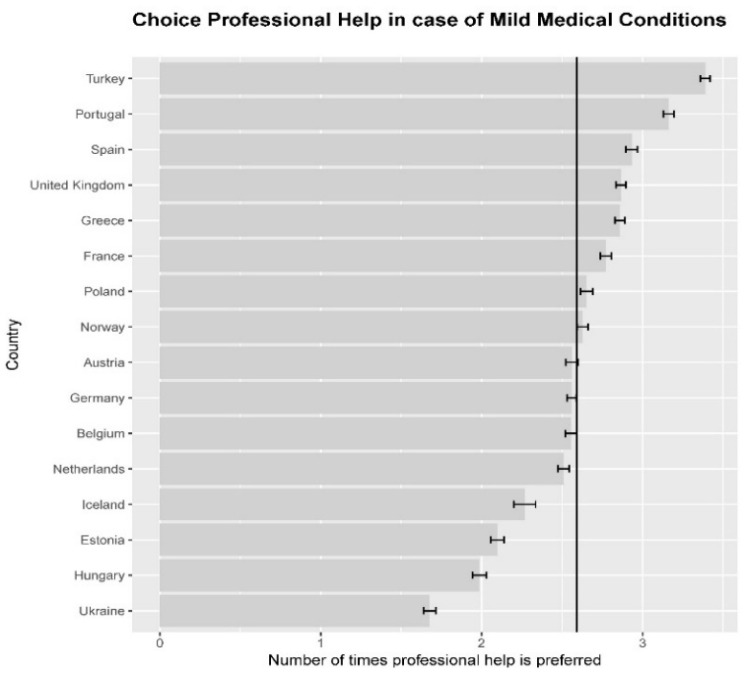
Preference for professional medical help in non-acute medical situations is a summated scale, showing preference or either professional help (1) or help from other sources (0) in the situation of serious headache, stomach-ache, back-ache or sleeping problems. Preference in these situations is summated into individual scores from 0 (preference for professional help in no situation) to 4 (always prefer professional help). The vertical line indicates the overall mean. For each nation the 5% confidence limits are shown; the graph indicates that there are substantial differences between nations in preference for professional medical help, with respondents from Turkey being the most prone to choose for professional help and respondents from Ukraine the least.

**Figure 3 ijerph-18-12808-f003:**
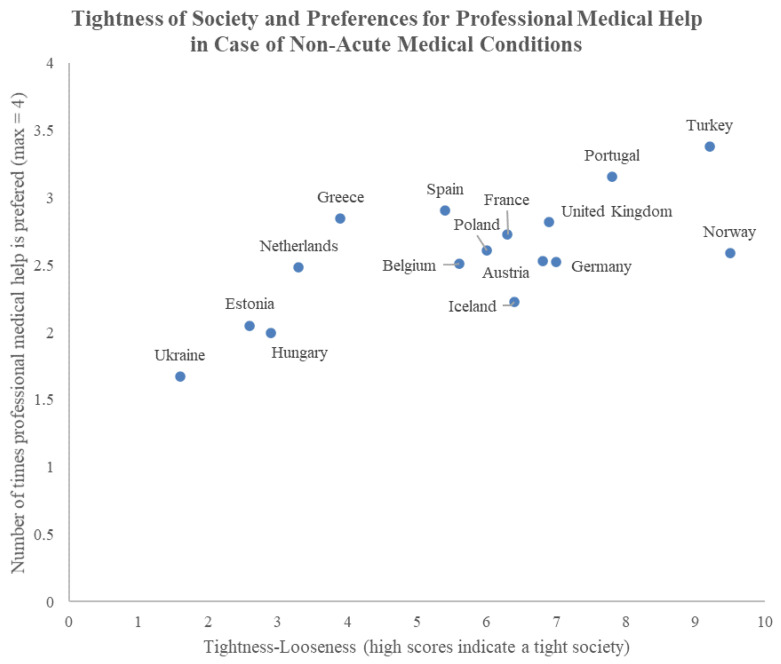
Relation between Tightness-Looseness and Preference for professional medical help in non-acute medical situations. The graph suggests a linear relation between Tightness-Looseness and the number of times professional medical help is preferred in the case of mild medical conditions (headache, stomach-ache, back-pain and sleeping problems). Tightness-Looseness is not related to GDP per capita as both Turkey (emerging economy) and Norway (highly developed economy) are both tight societies and Ukraine, a comparable economy to Turkey, has a low tightness.

**Table 1 ijerph-18-12808-t001:** Descriptive statistics for main variables.

Measurement Level	Parameter	Min	Max	M	SD
I-individual	Preference for professional medical help	0	4	2.64	1.415
I-individual	Age	18	99	46.85	17.479
I-individual	Education (in years)	0	32	11.44	4.314
I-individual	Health impairment	1	3	1.31	0.579
I-individual	Gender (Male = 1, Female = 0)	Female 54.6%	Male 45.4%	-	-
I-individual	Children in household (yes = 1, 0 = no)	0	1	0.24	0.426
I-individual	Perceived state of healthcare	0	10	4.95	2.609
I-individual	Self-perceived health	1	5	3.70	0.922
I-individual	Interpersonal trust	0	10	3.93	1.983
I-individual	Institutional trust	0	10	5.08	2.17
I-individual	Trust in doctor	1	6	2.51	0.682
I-individual	Conservation	−2.60	2.71	0.15	0.634
I-individual	Openness to change	−4.00	2.55	−0.25	0.640
I-individual	Self-enhancement	−3.52	2.10	−0.61	0.716
I-individual	Self-transcendence	−2.37	3.20	0.60	0.514
II-country	Tightness-looseness (TL)	1.60	9.50	5.79	2.197

**Table 2 ijerph-18-12808-t002:** Results multi-level models to predict preference for professional medical help in non-acute medical conditions, including variables at the individual level (indicated by I-) at the country level (indicated by II-) and cross-level interactions. Models 3–6 are nested in Model 2.

Measure Level	Parameter (γ)	Model 1	Model 2	Model 3	Model 4	Model 5
			(Conservation)	(Openness to Change)	(Self-Enhancement)	(Self-Transcendence)
I-individual	Intercept	2.695 ***	2.430 ***	2.431 ***	2.452 ***	2.492 ***
I-individual	Age centered	0.010 ***	0.008 ***	0.009 ***	0.010 ***	0.011 ***
I-individual	Age centr. and squared	−0.0002 ***	−0.0002 ***	−0.0002 ***	−0.0002 ***	−0.0002 ***
I-individual	Education centr.	−0.011 ***	−0.008 **	−0.010 ***	−0.011 ***	−0.011 ***
I-individual	Gender (M = 1, F = 0)	−0.128 ***	−0.114 ***	−0.121 ***	−0.125 ***	−0.143 ***
I-individual	Children < 12 in hh	0.031 **	0.023 *	0.026 *	0.031 **	0.032 **
I-individual	Health impaired	0.052 **	0.051 **	0.051 **	0.052 ***	0.054 **
I-individual	Self-perceived health	−0.050 ***	−0.044 **	−0.046 ***	−0.050 ***	−0.051 ***
I-individual	Perc. state of healthcare	−0.001	−0.002	−0.002	−0.001	−0.002
I-individual	Interpersonal trust	−0.019 ***	−0.016 **	−0.018 **	−0.019 ***	−0.017 **
I-individual	Institutional trust	0.029 ***	0.026 ***	0.027 ***	0.029 ***	0.029 ***
I-individual	Trust in medical doctor	0.065 ***	0.064 ***	0.065 ***	0.065 ***	0.064 ***
I-values	Conservation		0.102 ***			
	Openness to Change			−0.064 **		
	Self-Enhancement				0.007	
	Self-Transcendence					−0.049 *
II-country	Tightness-Looseness (TL)		0.133 ***	0.134 **	0.129 **	0.147 ***
Cross level	Conservation * TL		0.025 **			
	Openness to Change * TL			−0.006		
	Self-Enhancement * TL				−0.012 *	
	Self-Transcendence * TL					−0.021 **

***: Estimate is significant at the 0.001 level (2-tailed), **: Estimate is significant at the 0.01 level (2-tailed), *: Estimate is significant at the 0.05 level (2-tailed), *n* = 23,312.

## Data Availability

Individual Data for the paper is publicly available at the website of the European Social Survey (www.europeansocialsurvey.org (accessed on 29 November 2021)). Additional data has been collected from previous research and public resources [1,31,32,41,43] as described in the method section.

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
