# Peer review of "Health and Culture: The Association between Healthcare Preferences for Non-Acute Conditions, Human Values and Social Norms"

_ijerph, 2021, doi:10.3390/ijerph182312808_

Round 1
Reviewer 1 Report
Very interesting topic! I appreciate the use of the Schwartz theory to drive your methodology. I'm not sure that the results are all that surprising but it is helpful to have this validation! My other concern is that the countries are so different and varied in their health care approaches; I'm not sure how many would take your results and apply them, but again, it is interesting. Overall, I believe it is worthy of publication if some of these concerns are addressed..
Title: I would add "for Non-Acute Conditions" as that is the main emphasis of the paper.
Intro: First paragraph- my concern here is that the issue of "cost" as a driver for choice isn't mentioned; which, for some countries (e.g., USA) has a significant influence on where patients seek care.
Also missing from the intro...what are the costs in not seeking professional medical advice? And moreover, why is it important that we compare preferences across countries, given that the healthcare structures are different?
Line 67...wording "channel?"
The presentation of the Schwartz model is well done.
Prior to the methods, the "gap" in the literature needs to be better defined. Again, I would emphasize why we need to know this information?
Materials and Methods: line 166, replace the ; with a .
Also, please explain why other options besides doctor and nurse are given a 0, considering you are including pharmacists and other practitioners?
The methods section is otherwise well-detailed. That being said, my major issue is that cost of healthcare (given different structures) is not given a priority.
In the section on Robustness-checks (336-344)....does mention health insurance systems, but this needs more explanation.
Discussion: My comments are similar to above....the focus on tightness-looseness and other issues do not properly take into account issues such as cost or access, which are pretty important issues in some countries.
Conclusions: lines 407-408...our results may be of interest for governments, insurance companies, etc. Maybe...but so much of that is related to cost-benefit analysis.
Author Response
Dear reviewer 1,
Thank you very much for your positive review, and your critical comments. We tried to answer to your points as good as possible. Underneath you find our adaptations and remarks.
- Title: I would add "for Non-Acute Conditions" as that is the main emphasis of the paper.
Authors’ response:
Thank you for the suggestion, we changed it in the title
- Intro: First paragraph- my concern here is that the issue of "cost" as a driver for choice isn't mentioned; which, for some countries (e.g., USA) has a significant influence on where patients seek care.
Authors’ response:
We have addressed this point in the introduction by adding:
“Economic consequences of medical help seeking in the area of non-acute diseases can be are tremendous as prevalence of these diseases is high. Although in some countries this preference can be influenced by factors like accessibility and personal financial costs of the medical treatment, also other factors can play an important role.” Plus some additional rewriting of the first paragraph
- Also missing from the intro...what are the costs in not seeking professional medical advice? And moreover, why is it important that we compare preferences across countries, given that the healthcare structures are different?
Authors’ response:
Thanks for this remark: We addressed your first point by adding references on health costs and prevalence on headache, throat pain, back pain and sleeping problems.
As far as the second point is concerned: We acknowledge the differences between countries health systems and assessed the robustness of our findings. We estimated additional models including other country-level variables such as “type of health care insurance system”, “amount of physicians per 100.000”, and “GDP per capita”. Including these variables one by one we still find the same effects of our human values and tightness looseness constructs.
- Line 67...wording "channel?"
Authors’ response:
We changed the word to “option” to make the sentence more clear.
- The presentation of the Schwartz model is well done.
Authors’ response:
Thank you for your compliments.
- Prior to the methods, the "gap" in the literature needs to be better defined. Again, I would emphasize why we need to know this information?
Authors’ response:
To increase the clarity of the gap we added the following text to the end of the introduction: “Summarizing, research investigating human preferences for professional medical help is scant, and mostly limited to emergency room (mis)use. Research on seeking help in non-acute medical conditions is as far as we know lacking. Especially in non-acute healthcare conditions we expect cultural context and human values to have a prominent role in determining healthcare preferences.” to the end of the introduction.
- Materials and Methods: line 166, replace the ; with a .
Authors’ response:
We changed the sentence accordingly
- Also, please explain why other options besides doctor and nurse are given a 0, considering you are including pharmacists and other practitioners?
Authors’ response:
We added nurse to the choice for professional help (i.e., 1) as in some countries nurses seem have a more prominent role in giving care. We defined the difference based on payment for service: i.e., having a paid professional medical service versus all other non-paid options. We grouped pharmacist to the second option as this option could be seen as a way of self-help without paid consultation.
- The methods section is otherwise well-detailed. That being said, my major issue is that cost of healthcare (given different structures) is not given a priority.
Authors’ response:
We acknowledge that cost of healthcare to individuals can be an important predictor of preference, unfortunately we did not have a measure on the individual level that captured this cost of healthcare. To check the robustness of our model, we included several other country level measures that may capture preference for healthcare. A factor we considered was the difference in the way healthcare insurance was structured in the respective nations. In particular, we considered the amount of people with or without insurance and out-of-pocket healthcare expenditure, but in the context of the current sample we did not find any statistically significant effects.
- In the section on Robustness-checks (336-344)....does mention health insurance systems, but this needs more explanation.
Authors’ response:
To clarify this we added a sentence in this paragraph:
“we coded healthcare-insurance systems within countries on a 4 point scale using data from KPMG [43] ranging from fully publicly financed (1), mainly publicly financed (2), mixed financed (3) to mainly privately financed (4).”
- Discussion: My comments are similar to above....the focus on tightness-looseness and other issues do not properly take into account issues such as cost or access, which are pretty important issues in some countries.
Authors’ response:
We mentioned the out-of-pocket expenditure in the discussion, and added the your remark to the limitations.
- Conclusions: lines 407-408...our results may be of interest for governments, insurance companies, etc. Maybe...but so much of that is related to cost-benefit analysis.
Authors’ response:
We added in the conclusion: “As these common medical conditions are on the one hand related to increased cost of healthcare, and on the other hand to lost productivity and other social costs it is important to know that values add to explained variance in healthcare preferences.”
Reviewer 2 Report
This paper is evaluated as an important paper that expands the existing horizon in that it tries to examine the hypothesis in a different way than the previous approach by analyzing the attitude toward medical help in the treatment of non-acute medical conditions.
However, in the description of the manuscript, it should be corrected to use an abbreviation unfamiliar to researchers in other fields, refer to references without explaining important concepts clearly, or to be described in the research method are mixed with the research results.
To be clearly stated
- Description of the ESS used in the study
- Tightness-looseness, a key concept in proving the research hypothesis
- For researchers in other fields or researchers unfamiliar with analysis methods, it would be good to use the official terms written out of the abbreviated terms.
- It is also difficult to clearly understand the various models (Model3-Modle6) mentioned in the multi-level result section.
- Drawing simple regression line using Y and the values of tightness-looseness, which are integer rather than continuous in Figure 3 seems to be a description that goes beyond the intended purpose of transmission.
- It is difficult to understand it for the readers who do not fully understand the situation of the countries to be analyzed, to exclude the medical helpline or another practitioner from the medical help indicator for non-acute medical conditions.
- In preference for medical help, previous experience or availability of medical help may often be more important than the tightness-looseness. There is no mention of those parts in the robustness check of the Results.
Author Response
Dear reviewer 2,
Thank you very much for your positve comments and critical remarks. We tried to answer to your concerns as much as possible. I have attached the revised paper, and answered the questions below.
- However, in the description of the manuscript, it should be corrected to use an abbreviation unfamiliar to researchers in other fields, refer to references without explaining important concepts clearly, or to be described in the research method are mixed with the research results.
Authors’ response:
Thank you for your suggestion to write out the abbreviations; we completely agree with your observation and made appropriate changes. Also we rewrote the method section in order to improve clarity.
- To be clearly stated: Description of the ESS used in the study
Authors’ response:
We added a more detailed description of the European Social Survey (ESS) in the text, as well as a more clear reference to the European Social Survey in the abstract. (see attachment)
- To be clearly stated: Tightness-looseness, a key concept in proving the research hypothesis
Authors’ response:
We added to the abstract: “scores by Gelfand” (see attachment)
- For researchers in other fields or researchers unfamiliar with analysis methods, it would be good to use the official terms written out of the abbreviated terms.
Authors’ response:
We fully agree, we have added for each abbreviation also the fully written out terms in the paper. (see attachment)
- It is also difficult to clearly understand the various models (Model3-Modle6) mentioned in the multi-level result section.
Authors’ response:
We rewrote the results section and part of the discussion to improve readability and clarity. We removed model 1 as it did not add to our story. (see attachment)
- Drawing simple regression line using Y and the values of tightness-looseness, which are integer rather than continuous in Figure 3 seems to be a description that goes beyond the intended purpose of transmission.
Authors’ response:
We removed the regression line from the graph to avoid misunderstanding of the nature of the variable. (see attachment)
- It is difficult to understand it for the readers who do not fully understand the situation of the countries to be analyzed, to exclude the medical helpline or another practitioner from the medical help indicator for non-acute medical conditions.
Authors’ response:
We rewrote the description of the variable in line 175-189 to clarify the dependent variable. (see attachment)
- In preference for medical help, previous experience or availability of medical help may often be more important than the tightness-looseness. There is no mention of those parts in the robustness check of the Results.
Authors’ response:
Unfortunately previous experience of our respondents is not available in the data. As far as the second point is concerned: We acknowledge the differences between countries health systems and assessed the robustness of our findings. We estimated additional models including other country-level variables such as “type of health care insurance system”, “amount of physicians per 100.000”, and “GDP per capita”. Including these variables we still find the same effects of our human values and tightness looseness constructs.